# Alternative Splicing in Cancer and Immune Cells

**DOI:** 10.3390/cancers14071726

**Published:** 2022-03-28

**Authors:** Antoine Bernard, Romain Boidot, Frédérique Végran

**Affiliations:** 1Team CAdIR, CRI INSERM UMR1231 “Lipids, Nutrition and Cancer’’, 21000 Dijon, France; antoine.bernard.chum@ssss.gouv.qc.ca; 2Faculté des Sciences de Santé, Université Bourgogne Franche-Comté, 21000 Dijon, France; 3Molecular Biology Unit, Centre Georges-François Leclerc, UNICANCER, 21079 Dijon, France; rboidot@cgfl.fr; 4Institute de Chimie Moléculaire de l’Université de Bourgogne ICMUB, UMR CNRS 6302, 21000 Dijon, France

**Keywords:** alternative splicing, cancer cells, immune cells

## Abstract

**Simple Summary:**

Alternative splicing is one of the most fabulous and important mechanism in the cell. Alternative splicing is capable of generating many proteins from a single gene and can be involved in many pathways. In this review, we decided to present a review of the literature on alternative splicing in cancer cells but also in immune cells. This Review is composed of 4 different parts, with the impact of alternative splicing in cancer immunotherapy, the role of alternative splicing in immune modulation, the involvement of alternative splicing in cancer cells and finally, the cause of deregulation of alternative splicing in cancer.

**Abstract:**

Splicing is a phenomenon enabling the excision of introns from pre-mRNA to give rise to mature mRNA. All the 20,000 genes of the human genome are concerned by this mechanism. Nevertheless, it is estimated that the proteome is composed of more than 100,000 proteins. How to go from 20,000 genes to more than 100,000 proteins? Alternative splicing (AS) is in charge of this diversity of proteins. AS which is found in most of the cells of an organism, participates in normal cells and in particular in immune cells, in the regulation of cellular behavior. In cancer, AS is highly dysregulated and involved in almost all of the hallmarks that characterize tumor cells. In view of the close link that exists between tumors and the immune system, we present in this review the literature relating to alternative splicing and immunotherapy. We also provide a global but not exhaustive view of AS in the immune system and tumor cells linked to the events that can lead to AS dysregulation in tumors.

## 1. Introduction

It has long been estimated that the human genome was composed of nearly 300,000 genes. With the advent of new sequencing technologies, it is now assumed that we hold around 20,000 genes. This state of affairs may seem surprising at first. Indeed, many species such as zebrafish or the nematode worm hold as many genes or more. Thus, there would be a paradox that the number of genes is not representative of the biological complexity of an organism. Several mechanisms that can address this problem have been discovered, including alternative splicing (AS) of pre-messenger RNAs. This phenomenon makes it possible to obtain from a single gene several functional proteins. If at the time of the discovery of AS it was estimated that about 10 to 15% of human genes produced multiple transcripts, it is now established by RNA sequencing that it is actually 95 to 98% of genes that generate multiple isoforms. AS which puts an end to the “a gene, a protein” paradigm leads to a reconsideration of the very conception of what a gene is.

If AS grants healthy cells flexibility and adaptability according to a given environment or stimulus, it can sometimes be deleterious and contribute to their transformation into cancer cells. Indeed, more and more studies show that AS has a preponderant role in cancer development. On the other hand, it has been shown that AS is involved in the homeostasis and differentiation of immune cells and especially those of acquired immunity. Through the different aggression of non-self to which they must respond, these cells are brought to evolve and adapt in as many different environments as a ganglion, blood, or lymphatic circulation or even a tumor can be. Thus, the study of the AS of these cells could bring a new level of understanding of the behavior of these cells according to a given environment. The elucidation of these mechanisms could allow the creation of tools to exacerbate or otherwise inhibit certain immune reactions depending on a given pathophysiological situation or in the context of treatment such as anti-tumor immunotherapies.

## 2. Alternative Splicing and Cancer Immunotherapy

Immunotherapies are therapeutic strategies very effective in around 15–20% of patients. The presentation of tumor neoantigens seems to be one of the predictive markers of immunotherapies response (Figure 1).

### 2.1. Alternative Splicing Can Increase Immunotherapy Response

Immunotherapy targeting immune checkpoints such as PD-1 (programmed cell death 1) or CTLA-4 (cytotoxic T lymphocyte-associated protein 4) is a real revolution in the treatment of many cancers [1]. In this context, it has been observed that tumors with a high mutational load have more tumor peptides at their disposal surface [2]. However, while many studies show a positive correlation between the level of tumor mutation and the response to checkpoint inhibitors [3], others contradict them by showing that sometimes highly mutated patients do not respond to treatment, while other patients with few mutations respond to treatment [4]. In this context, it has been shown that patients with somatic mutations generating new splice sites respond better to immunotherapies targeting PD-L1 [5]. This type of mutation could then be a new clinical biomarker of choice in the response to checkpoint inhibitors. New technologies such as RNA sequencing or mass spectrometry (MS) now make it possible to identify tumor neo-antigens.

With the advent of cell therapy techniques, it is now possible to target these antigens by selecting a cytotoxic lymphocyte clone specific for a given tumor antigen, then amplifying it and performing an adoptive transfer of these cells. It is also conceivable to use cytotoxic lymphocytes that have been reprogrammed genetically to express a chimeric TCR that makes it possible to recognize a specific tumor antigen (CAR-T cells (chimeric antigen receptor T cells). Nevertheless, there are still many obstacles to the implementation of therapeutic strategies to specifically target these antigens. It is first of all crucial to determine whether the tumor neoantigen is specific to the tumor.

### 2.2. Alternative Splicing Generates Neoantigens

It has recently been demonstrated in a study of more than 8705 patients, that tumors had more than 30% more AS events than healthy tissue samples [6]. Focusing on ovarian and breast tumors that also had MS data and transcriptomic data, 68% of tumors were found to have one or more tumor neo-epitopes derived from the tumor. In contrast, only 30% of tumors have tumor neo-epitopes from somatic mutations. Thus, it highlights the importance of analyzing AS events within tumors to identify new targets. Moreover, another study using MS demonstrated that most tumor antigens from two murine lines and seven primary human tumors originated from the translation of coding exons out of the reading frame, but also from non-coding regions of the DNA. All of these events can most likely originate from the deregulation of AS in these tumors [7].

Since AS is tissue-specific regulated, it is essential to analyze whether the AS event found within the tumor does not occur elsewhere in the body and not just in the healthy tissue surrounding the tumor [8]. The transcriptomic databases available for each organ could answer this problem in order to avoid any nonspecific immune responses that could have dramatic effects. In addition, it is still difficult to determine AS events at the subclonal level in tumors. Single-cell RNA sequencing may be the key to overcoming this technical difficulty, but the starting material used for this technique is currently too weak to expect anything other than the expression of highly represented transcripts in cells [9]. However, new algorithms appear promising in identifying alternative transcripts from relatively low coverage RNA sequencing [10]. Regarding MS, other problems come to oppose the use of tumor neo-antigens from AS. Indeed, the detection threshold is often very high and requires a large quantity of cells to detect minor events of AS at the protein level [11]. In addition, it is important to test the immunogenicity of tumor neo-epitopes from AS. Indeed, in melanoma and glioblastoma, 51.7–66% of neo-epitopes presented by class II MHC generate a CD4 + T cell response, and only 16–43% of those presented by class I induce a response from the cytotoxic lymphocytes. Factors such as the specificity of peptides for MHC, their abundance but also the effectiveness of their presentation should be taken into account. Nevertheless, it appears that many neo-epitopes derived from AS have an immunogenic capacity [12,13,14]. Huge progress has been made in cancer immunotherapies over the past decade [15]. Neo-epitopes from tumor-related AS are slowly being considered to develop promising new cell therapies for patients.

### 2.3. Alternative Splicing Can Limit Inhibitor of Checkpoint Inhibitor Response

AS may also limit the effects of immune checkpoint inhibitor-based immunotherapies. Recently, a PD-L1 splice variant has been identified in several cancers. It appears to be secreted and accumulates in the tumor microenvironment [16]. This variant has been shown to be involved in anti-PD-L1 resistance. CTLA-4 is another checkpoint inhibitor that has two isoforms: a membrane mCTLA-4 and a secreted form sCTLA-4. sCTLA-4 is a splicing variant devoid of exon 3 [17]. Studies have shown that both variants are expressed in tumor cells and are associated with immune escape from tumors [18,19]. Both bind to CD80 and CD86 on antigen-presenting cells (APCs) inducing a feedback of T cell activation [20]. At present, both variants have shown immunosuppressive activity but their prognostic role remains to be investigated.

### 2.4. Alternative Splicing May Limit the Efficiency of CAR-T Cells

The CD19 antigen is expressed on the surface of the majority of B cells in acute lymphoblastic leukemia. Cell therapy using chimeric TCR T cells specifically recognizing this antigen gives very good results. However, in 10 to 20% of cases, a relapse is noted after this treatment due to a decrease in the expression of this epitope [21]. It has recently been discovered that most of these relapses were due to a CD19 AS [22]. Since the alternative isoform CD19Δex2 is not recognized by the CAR-T-19, this could explain the relapses observed in some patients. The modification of the AS of this transcript is under the influence of the splicing factor SRSF3, but the activation of the latter during the relapse remains misunderstood. However, this discovery has isolated a new target from AS that will prevent many relapses in people with this disease and treated with CAR-T cells.

## 3. Alternative Splicing in Immunity Modulation

In the immune system, alternative isoforms have been identified in physiological contexts and others in pathological contexts, both in the innate and acquired immune systems [23] (Figure 2). For example, in macrophages, 81 AS events were observed during LPS stimulation [24]. Dendritic cells also develop different AS patterns, depending on the status of their differentiation and maturity. Single-cell RNA-sequencing has shown that different AS patterns exist depending on the maturity and differentiation of dendritic cells exposed to LPS [25]. However, the functional impact of these different splicing profiles has not been elucidated [25]. For T lymphocytes, as many as 1319 AS events occur during TCR stimulation and 1575 during TCR commitment and CD28 costimulation [26], including genes involved in the immune response [27]. Finally, more than 90% of B-cell multi-exon genes undergo AS [28]. Nevertheless, the role of alternative isoforms in immunity remains unclear. Moreover, the role of AS has not yet been characterized in all cell types and even less for tumor-infiltrating cells. The complexity of this phenomenon can be highlighted by the few examples that we present below.

### 3.1. Immunoglobulins and B Cells

B cells are important players in adaptive immunity, by their propensity to secrete significant amounts of antibodies directed against a multitude of pathogens. In order to recognize many antigens, B cells rearrange the variable portion of their heavy and light chains within the bone marrow. These immature cells express on their surface IgM (immunoglobulin M). Once tested for their ability to respond to the self, conforming cells migrate to secondary lymphoid organs such as the spleen or lymph nodes. It is then the IgD that is expressed on the membrane surface. The reason for this transition from IgM to IgD has not yet been fully elucidated. However, it was observed that AS was responsible for the transition from the IgM isotype to the IgD isotype. The pre-mRNA encoding the IgM heavy chain contains the variable domain of recognition of VDJ antigens, the μ domain, but also the sequence encoding IgDs (δ domain). Upon the departure of B lymphocytes to secondary lymphoid organs, the AS that formerly led to the expression of IgM is directed towards a concomitant production of the two isotypes [29]. Although the exact role of this change is not yet fully known, it is known that the protein binding to the Zfp218 (zinc finger protein 318) RNA could be one of the players in the regulation of this AS. In fact, in follicular B cells, whose response depends on T lymphocytes, Zfp318 increases the amount of IgD at the expense of IgM [30]. Depending on their state of maturity, B cells are also able to secrete IgG, IgA, and IgE. It has been noted that IgE has several isoforms that appear according to different stimuli and the level of maturity of B cells [31]. AS is important in establishing the immune response by regulating its activation but also by orienting its differentiation.

### 3.2. Immune Cell Receptors

The importance of AS in macrophages has been demonstrated through the inhibition of the SF3A1 splicing factor that regulates the splicing of genes involved in the signaling pathway under TLRs [24]. Indeed, this leads to a decrease in the production of important regulators of this pathway, such as CD14, IRAK1 (interleukin-1 receptor-associated kinase 1), or IKKβ (inhibitor of nuclear factor kappa-B kinase subunit beta). This shows the importance of AS in the propensity of the innate system to activate in contact with the non-self through TLRs. It has also been demonstrated in macrophages and dendritic cells that there is a soluble form of TLR4 with a 144 base pair insertion between exon 2 and 3 leading to the appearance of a premature STOP codon. The appearance of this alternative isoform of TLR4 occurs during LPS stimulation. This results in decreased secretion of TNF-α and decreased activation of NFκB [32].

MyD88 (myeloid differentiation primary response 88) is a protein of great importance in the signaling pathway under TLRs. There is a long isoform of MyD88, MyD88L, that is responsible for the activation of innate immunity and a shorter isoform MyD88S, which instead restricts the immune response by blocking the signaling pathway. It was observed in this context that SF3A1, SF3A2, SF3A3, and SF3B1 splice factors were essential for the formation of the long form of MyD88 and that their genetic invalidation caused an inhibition of the innate immune response [33]. Moreover, these observations are valid in both macrophages and dendritic cells.

The IL-33 ST2 receptor is expressed on immune cells such as Th2, Treg, or even ILC2. AS of ST2 generates three isoforms: a transmembrane isoform named ST2L, a soluble isoform sST2, and ST2V [34]. sST2 acts as a “bait receptor” to competitively bind to IL-33 and block activation of the IL-33/ST2 signal [35,36] which plays an important role in the development of colorectal cancer [37].

The NLRP3 protein (NOD-like receptor family, pyrin domain containing 3) is involved in the NLRP3 inflammasome also composed of ASC (apoptosis-associated Speck-like protein containing CARD) and caspase-1. Largely expressed in macrophages, NLRP3 belongs to the PRR family (pattern recognition receptor), receptors for detecting molecules associated with pathogens. Upon activation, NLRP3 inflammasome cleaves pro-IL-1β and pro-IL-18 into IL-1β and IL-18, pro-inflammatory cytokines. NLRP3 is composed of three functional domains: PYD (N-terminal pyrin domain), the NACHT domain, and the LRR domain (leucine-rich repeat). LLR domain is responsible for inflammasome formation. A recent study has shown the existence of several alternative isoforms of NLRP3 in humans, losing either exon 5 or exon 7 or both exons [38]. The exclusion of exon 5 directly affects the LRR domain of NLRP3 that is involved in protein-protein interactions. As a result, the LRRΔE5 domain no longer interacts with the NEK7 protein (NIMA (never in mitosis A)—related kinase 7) [39]. As a result, caspase 1 is no longer able to convert pro-IL-1β and pro-IL-18. Thus, using antisense oligonucleotides, it would be possible to direct AS towards exon 5 exclusion in order to limit the onset of inflammation [38].

One of the first genes undergoing an AS to be identified in lymphocytes was Ptprc which encodes the CD45 protein. This tyrosine phosphatase is found in abundance on the surface of T and B lymphocytes, representing about 10% of total membrane proteins. CD45 is known to regulate TCR signaling by dephosphorylating Lck kinase. The pre-mRNA of Ptprc has three variable exons (4, 5, and 6) that generate the CD45RA, RB, RC, RO isoforms. In naïve T cells, CD45 is expressed either with variable exon 4 or with variable exon 5, namely CD45RA and CD45RB. In contrast, it was observed that in memory T cells all exons were excluded to form CD45RO [40]. There are functional differences between CD45 isoforms. Isoforms with a variable exon decrease the TCR activation threshold making cells more easily to be activated. The CD45RO isoform without variable exons attenuates TCR activation [41]. CD44 plays an important role in the migration of lymphocytes, in particular by allowing the phenomenon of rolling adherence and then firm adhesion with vascular endothelial cells expressing HA [42]. Alternative isoforms of CD44 have been observed in activated lymphocytes. The most expressed of these includes variable exon 5 (see CD44 part in cancer), and its appearance depends on the Sam68 RNA binding protein [43]. Sam68 has been shown to be activated by the Ras/Raf/MEK/ERK signaling cascade following the commitment of the TCR. Once active, Sam68 forms a complex with the SR SRm160 protein that interacts with the ESE sequence of exon 5 that induces its inclusion. This interaction with the CD44 pre-mRNA facilitates spliceosome recognition of splice sites. It has also been shown that by associating with the Brm protein, Sam68 is able to slow down the progression of RNA polymerase II that induces the inclusion of exon 5. Low splice sites are less attractive through their sequence for the spliceosome that prefers to attach to strong splice sites. This slowdown thus allows the spliceosome players to attach to the so-called weak splice sites surrounding exon 5. For the moment, little is known about the potential action of this alternative isoform in the immune response. Nevertheless, it is known that this isoform including exon 5 is more expressed in some autoimmune diseases [44].

### 3.3. Transcription Factors

The transcription factor Foxp3 is of great importance in regulatory T cells (Treg). In total, three isoforms of Foxp3 have been described: a full-length isoform Foxp3fl, another losing exon 2 Foxp3Δ2, and finally a last losing both exon 2 and exon 7, Foxp3Δ2Δ7. The different isoforms of Foxp3 do not all have the same effect on Treg differentiation. Indeed, while the first two maintain the immunosuppressive character of Treg lymphocytes, the latter does not. It has also been shown that when Foxp3fl and Foxp3Δ2Δ7 were coexpressed in comparable proportions, Foxp3Δ2Δ7 acts as a dominant negative and inhibits the activity of Foxp3fl [45]. It was also observed that exon 7 is of great importance for Foxp3. In individuals mutated at the donor splice site of intron 7, exon 7 is excluded. This results in a severe disruption of the immune system leading to IPEX syndrome (Immunodysregulation polyendocrinopathy enteropathy X-linked) manifested by polyendocrinopathy and autoimmune enteropathy [46]. Another study also described the impact of AS of the Foxp3 transcription factor in the plasticity of Treg cells. They first observed a significant increase in the Foxp3Δ2Δ7 alternative isoform in peripheral blood mononuclear cells (PBMC) and in intestinal biopsies of patients with Crohn’s disease compared to healthy patients. They then found that Tregs exposed to the pro-inflammatory cytokine IL-1β express Foxp3Δ2Δ7. However, the mechanism explaining the increased exclusion of the two Foxp3 variable exons is not yet explained. It is possible that this is due to the activation of splicing factors in the signaling pathway under the IL-1β receptor. Cells that express more this alternative isoform secrete more IL-17a, the cytokine characteristic of Th17 differentiation [45]. Indeed, this alternative isoform, unlike Foxp3fl, is not able to bind to the transcription factor RORγt to prevent it from inducing transcription of the gene encoding IL-17a. Thus, the Foxp3Δ2Δ7 isoform has a double effect in Crohn’s disease: inflammation is increased by the decrease in the Treg population but also by the increase in the Th17 population largely implicated in this disease [47].

Although Tbet is the key transcription factor in the differentiation of CD4 T subtype Th1, other transcription factors have been shown to be essential during the differentiation process. In Th1 cells, expression of IRF1 depends on either STAT1 via IFNγ or STAT4 via IL-12 pathways. This transcription factor has been described as essential for the expression of the IL-12 receptor β1 subunit [48,49]. Indeed, by acting in cooperation with Tbet, IRF1 allows the expression of a functional receptor provided with β1 and β2 subunits. This results in the amplification of the IL-12 cytokine signaling which supports Th1 lymphocyte orientation. There is an AS of Irf1 pre-mRNA in Th1 cells in vitro and in vivo under the influence of TGF-β. In fact, a short form of IRF1, lacking exon 7, appears during Th1 differentiation and its expression is increased in tumor settings and during exposure to TGF-β. A repressive action of IRF1-short in Th1 lymphocyte differentiation has been described. By interacting with the complete isoform of IRF1, IRF1-short prevents its binding to the *Il12rb1* promoter [50].

### 3.4. Immune Checkpoints

PD-1 is encoded by the PDCD1 gene. The main PDCD1 AS event is exon 3 skipping, which results in the formation of the PD-1Δ3 isoform. PD-1Δ3 is a soluble form of PD-1 which prevents the interaction between PD-L1 and PD-1. Clinical studies have shown a correlation between prognosis and expression of PD-1Δ3 in patients with non-small lung cancer during treatment with erlotinib [51]. An antisense oligonucleotide targeting strategy can modify PDCD1 splicing and favor the generation of PD-1Δ3 to exert a therapeutic effect [52].

CTLA4 is another immune checkpoint that also has splice isoforms. Soluble CTLA4 (sCTLA4) inhibits the interaction between B7 and CD28 by binding to B7on antigen presenting cells. It thus inhibits the activation of T cells. Anti-sCTLA4 monoclonal antibodies bind specifically to sCTLA4 but do not recognize CTLA4, thus enhancing the T cell-specific response and exerting anti-tumor activity [53].

## 4. Alternative Splicing in Cancer Cells 

Under physiological conditions, AS causes the increase in genetic diversity by obtaining several transcripts and therefore several proteins from a single gene. Nevertheless, if this phenomenon grants healthy cell flexibility and adaptability, it can sometimes be deleterious and contributes to their transformation into cancer cells through affecting proliferation, apoptosis, invasion, migration, or metabolism (Figure 3). More and more studies show that AS has a leading role in cancer development and progression [54,55,56,57]. Here, we present the flagship cellular functions modulated by AS.

### 4.1. Alternative Splicing of Tumor Suppressors

The tumor suppressor P53 (or TP53 tumor protein 53) is of major importance. It is involved in numerous cellular processes such as cell cycle regulation, or regulation of many genes involved in the cell cycle, autophagy, metabolism, and apoptosis. In more than 50% of cancers, the gene coding for p53 is mutated, which annihilates its anti-proliferative and apoptotic abilities. However, until recently, studies around P53 focused only on the main transcript of this gene and not on the different transcripts and their potentially variable activities. Thus, it has been shown that P53 can be mutated at intronic splice sites giving rise to truncated proteins just as inactive as for a somatic mutation. When a splice site is affected, aberrant transcripts of P53 are produced and the genes usually under its influence are reduced in expression [58]. There are in total nine known alternative isoforms of this transcription factor. These isoforms are expressed in a tissue-specific manner and may explain why some react differently to stress such as ionizing radiation [59,60]. The ΔP53 variant (44KDa) lost 66 amino acid residues mostly located in the DNA binding domain. The tetramerization domain is intact which allows it to homo-oligomerize. Despite the alteration of the DNA binding domain, Rohaly et al. showed, by chromatin immunoprecipitation, that ΔP53 was capable of binding, like P53, the promoters of the *p21* and 14-3-3 genes, involved in the cell cycle. Unlike P53, ΔP53 does not bind to the promoters of *mdm2*, *bax*, and *pig3*, involved in apoptosis. This shows that this protein is probably more related to the cell cycle than to apoptosis [61]. The ΔP53 protein is activated by phosphorylation of Ser15 by ATR, at the beginning of the S phase, in response to UV irradiation during the G1/S transition. This induces a transcriptional activity of ΔP53 resulting in the expression of the *p21* gene at the beginning of the S phase. In the absence of ΔP53, the checkpoint of the S phase does not exist and is replaced by the G2 phase. The ΔP53 protein is therefore an essential player in the ATR-intra S phase checkpoint [61]. The P47 variant (47KDa) has completely lost the transactivation domain relative to P53. However, via its oligomerization domain, it is able to bind to P53. Upon overexpression of P47 and adriamycin-induced stress, P47 relocates from the nucleus to the cytoplasm. This phenomenon is associated with the nuclear export of P53. This confirms the interaction observed by immunoprecipitation between the two proteins. In addition, P47 promotes the mono-ubiquitination of P53, which leads to the nuclear export of P53 as well as Mdm2 [62]. The protein Δ133P53, with a molecular weight of 35 KDa, does not contain a transactivation domain and has a DNA binding domain truncated in its N-terminal part. To date, only its influence on apoptosis has been explored. It appears that when H1299 lung cancer cells are co-transfected with P53 and Δ133P53, P53-dependent apoptosis is greatly reduced, suggesting Δ133P53 is a dominant negative of P53 [63]. The α and β isoforms are truncated proteins ending respectively in 10 and 15 amino acids downstream of exon 8. These isoforms, therefore, do not have an oligomerization domain. Only the role of P53β has been explored. First, it turns out that P53β is capable of forming a protein complex with P53. Then, it was shown that P53β binds preferentially to the promoter of *Bax* and *p21*, while P53 binds little to the promoter of *bax* compared to those of *p21* and *mdm2*. This, therefore, suggests that P53β might affect the transcriptional activity of P53 on the *bax* promoter. Indeed, in the absence of P53β, P53 binds poorly to the *bax* promoter, whereas in the presence of P53β, P53 increases the transcription of *bax* without increasing the level of P53 protein. In terms of the role in apoptosis, the presence of P53β does not influence the apoptosis induced by P53 [63].

The IRF1 transcription factor (interferon regulatory factor 1) has been identified for its ability to induce expression of the gene encoding IFN-γ [64]. Afterwards, it has been described for its role in cell cycle arrest following DNA damage [65]. Depending on the cell type, IRF1 is able to act alone but also in cooperation with P53 [66]. Thus, IRF1 is a tumor suppressor and also plays a role in the immune response. In cancer cells, IRF1 is a tumor suppressor. However, in many cancers, the functions of IRF1 in cell death and in cell cycle regulation are impaired. In cervical cancer, it has been shown that HPV-16 and 18 (human papillomavirus) E7 oncoprotein can repress IRF1-regulated transcriptional regulations by binding to the latter [67]. In leukemic cells, two alternative isoforms of IRF1 losing exon 2 or exons 2 and 3 have been identified. Nevertheless, since these isoforms also lose their DNA binding domain, they cannot exert their role as a transcription factor and their presence does not seem to disturb the activity of the standard form of IRF1 [68]. However, this time in cervical cancer, four alternative transcripts were identified as being more expressed in cancer tissue than in healthy tissue. Through experiments using cloning strategies, it was observed that these isoforms had repressive activity on the transcriptional activity of the standard form of IRF1. These alternative isoforms also seem more stable over time. Since IRF1 is responsible for regulating the expression of tumor suppressor genes (*p73*, *FHIT*, *TGS101*), these isoforms aggravate the proliferation and aggressiveness of this type of cancer [69].

The apoptosis-stimulating protein of P53-2 (ASPP2) is a tumor suppressor promoting apoptosis mediated by P53 through direct interaction with P53. An oncogenic splicing variant of ASPP2, with a high prevalence in acute leukemia, has been identified. This isoform (ASPP2K) has a truncated C-terminal domain, losing the P53 binding sites. Thus, ASPP2K possesses dominant-negative activity impairing the induction of P53 dependent apoptosis. ASPP2K is expressed in CD34+ leukemic progenitor cells and functional studies suggest a role in early oncogenesis, causing in addition to impaired induction of apoptosis, dysregulation of proliferation similar to P53 mutations. ASPP2K variant makes cancer cells more aggressive [70].

### 4.2. Alternative Splicing Modulates Proliferation

An alteration in the regulation of the different steps of the cell cycle results in an anarchic proliferation of cancer cells. In this context, a number of genes involved in cell cycle regulation exhibit splice variants. For example, the gene CCND1 encodes two splice isoforms of cyclin D1: cyclin D1a and cyclin D1b [71]. The formation of cyclin D1b is directly linked to a G/A polymorphism at the exon 4 and intron 4 junctions of the gene [72]. Cyclin D1a variants interact with CDK4/6 to form a complex and are responsible for the phosphorylation of the tumor suppressor protein Rb. The latter promotes cell cycle progression and consequently proliferation of cells. Cyclin D1b can also interact with CDK4 but is not capable to phosphorylate Rb [73,74]. Thus, there may be a balance between cyclin D1b and cyclin D1a. The first limits cancer growth by opposing the functions of cyclin D1a [75].

### 4.3. Alternative Splicing of Apoptosis Related Genes

Resistance to cell death and apoptosis was early identified as a “Hallmark of cancer” [76]. Apoptosis is a natural barrier to the development of cancers within an organism. The apoptotic machinery is made up of a very large number of actors harboring either pro- or anti-apoptotic properties allowing for the precise regulation of this phenomenon. Numerous dysregulations of the mechanisms regulating apoptosis occur during carcinogenesis and appear to participate in resistance to chemotherapy treatments [77]. It has in particular been shown that dysregulation of AS would promote the expression of variants possessing anti-apoptotic activity.

Caspases are proteins that play an essential role during apoptosis. One of these caspases, caspase-3, is a pro-apoptotic protein that is activated from a proform by cleavage during apoptosis [78]. It possesses a catalytic domain that allows its function during apoptosis, in particular, induced why chemotherapeutic agents. AS of caspase-3 gives rise to one alternative isoform, caspase-3s [79]. This variant is significantly more expressed in cancers than in healthy tissues and does not have a catalytic site. This isoform inhibits apoptosis. Indeed, caspase-3s can interact with pro-caspase-3, impairing apoptosome assembly and caspase-3 activation [80]. Nevertheless, targeting splicing of caspase-3 could be difficult in cancer treatment as the main isoform of caspase-3 was also described as a regulator of angiogenesis promoting chemotherapy resistance [81].

Bcl-2 (B-cell lymphoma 2), a target of survivin-ΔEx3, is a protein involved in the regulation of cell death. In particular, it allows, in combination with other proteins of its family such as Bax (Bcl-2-associated X protein) and Bak (Bcl-2 homologous antagonist/killer), to regulate mitochondrial permeability. By allowing the release of cytochrome c and ROS, Bcl-2 induces a signaling cascade that can lead to cell apoptosis. There is an apoptotic version of Bcl-2 called Bcl-XS, and an anti-apoptotic version called Bcl-XL. The latter results from the exclusion of exon 2 during splicing. This AS phenomenon is increased in many cancers and leads to resistance to apoptosis usually induced by chemotherapy used to treat them. Treatments are currently being developed around the use of oligonucleotides carried by lipid nanoparticles. The results are conclusive in models of pulmonary metastases using B16F10 cancer cells [82].

Anti-apoptotic actors such as survivin also undergo AS. Survivin was identified in 1997 [83]. In a physiological context, this protein is weakly expressed and its overexpression has been observed in most human cancers. For a long time, five isoforms have been described and studied but recent data obtained with high throughput sequencing revealed 14 new isoforms [84].

Among the different alternative variants that were studied, survivin-ΔEx3 keeps anti-apoptotic functions by binding caspase-3 and Bcl-2 to block their cleavage [85]. Two other described isoforms, survivin-2B [86] and survivin-2alpha [87], were described as pro-apoptotic isoforms. Finally, the last one, survivin-3B, is specifically expressed in tumors and harbors anti-apoptotic properties. It is able to bind caspase-8 and caspase-6 to promote immune escape and resistance to chemotherapy [88,89]. Survivin has become a real therapeutic target but targeting one of its isoforms, especially survivin-3B, could be useful for cancer treatment.

### 4.4. Alternative Splicing and Angiogenesis

The provision of nutrients and oxygen during neoplastic expansion requires the establishment of a new vascular network. VEGF-A is the most prominent angiogenic factor among members of the VEGF family. It is of major importance, among others, in wound healing, embryonic development, but also in cardiovascular diseases and cancer [90]. The gene encoding VEGF-A is composed of eight exons and it is now established that splicing events occur in exons 5, 6, 7, and 8. Nevertheless, the two major isoforms of VEGF-A are VEGF-xxxa and VEGF-xxxb. The first one is pro-angiogenic and the second one is anti-angiogenic. The expression of VEGF-xxxb is often decreased in cancers. This results in increased angiogenesis that accelerates tumor growth [91].

The biological effect of VEGFA occurs following its interaction with its receptor (VEGFR) at the surface of cells (endothelial cells, macrophages, neutrophils, etc.). It has a soluble splicing isoform called sVEGFR2 that does not affect angiogenesis but inhibits the proliferation of endothelial cells, especially in human lymphangioma [92]. Regarding VEGFR1, to date, four splice variants have been identified among which, sVEGFR1-113 is considered to be a natural antagonist of VEGFA [93].

### 4.5. Alternative Splicing and Metabolism Regulation

There is growing evidence that cancer-related AS helps to promote tumor growth and proliferation, in part due to the effects of metabolic reprogramming. In many types of cancer, AS acts as a molecular switch to alter metabolism and stimulate tumorigenesis.

Compared to normal cells, tumor cells preferentially metabolize glucose by aerobic glycolysis. Glycolysis consists of a series of reactions catalyzed by enzymes coded by genes, some of which are subject to AS. Pyruvate kinase, for example, catalyzes the last step of glycolysis and converts phosphoenolpyruvate to pyruvate. The gene encoding muscle isozyme pyruvate kinase *PKM*, generates isoforms PKM1 and PKM2 by AS using exon 9 and exon 10, respectively. PKM1 is found in most normal cells. It promotes oxidative phosphorylation and mitochondrial metabolism pathway. Conversely, the PKM2 isoform is upregulated in tumor cells and promotes aerobic glycolysis [94].

Changes in fatty acid metabolism have also been identified as another important metabolic abnormality during tumor progression [95]. Fatty acids play an essential role in energy storage, membrane synthesis, and the production of signal molecules [96]. For this, fatty acids must be converted into acyl-coenzyme A, which will be used later for subsequent metabolism (anabolism or catabolism). Acyl-coenzyme A synthetase catalyzes the conversion of fatty acids [97]. Members of the long-chain acyl-CoA synthetase (ACSL) family include five isoforms each encoded by a distinct gene and having several splice variants [98]. The ACSL4 variant is significantly more expressed in hepatic tumor cells than in normal tissues. This protein participates in tumorigenesis by disrupting lipid metabolic pathways. ACSL4 expression has also been observed in breast cancer cells promoting tumor growth.

### 4.6. Alternative Splicing and Epithelial-Mesenchymal Transformation

The epithelial-mesenchymal transformation (EMT) is considered to be the key mechanism underlying the metastatic and invasive capacities of malignant tumors derived from epithelial tissue [99]. Although a 2011 study showed there was a specific transcriptomic signature of AS during EMT [100], only CD44 and its variants have been actually studied.

CD44 is a transmembrane receptor involved in cell proliferation, invasion, and metastasis. CD44 has dozens of splice variants. Different isoforms of CD44 are expressed according to the type of cancer, and cancers with different isoforms have been shown to be more aggressive than those with only CD44s [101]. In the murine 4T1 breast cancer cell line, cells expressing the CD44v8-10 alternative isoform including the variable portion of exon 8 to 10 were found to have a higher metastatic potential than those only expressing the standard form of CD44. Moreover, the expression of different isoforms of CD44 can confer a stem cell profile to tumor cells of many cancers. These tumor cells can give rise to several types of tumor cells, making them highly tumorigenic and turning the tumor more difficult to eliminate [102]. Another variant including variable exon 6 (CD44v6) has been identified as being responsible for the development of metastases in colorectal carcinoma [103]. It has also been noted that this variant is involved in the epithelial-mesenchymal transition in prostate cancer [104]. In some cancers such as squamous cell carcinoma of the head and neck, the disappearance of the isoform CD44v6 is associated with a poorer prognosis. In addition, other studies show that it is the standard form of CD44 that is involved in the emergence of EMT [105]. The explanation could come from hyaluronic acid (HA), which interacts with all isoforms of CD44, and seems to be a universal factor in the establishment of EMT. The differences thus observed according to the type of tissue or cancer would come from the amount of HA or from the specificity of the different isoforms of CD44 for this molecule [106]. Nevertheless, the regulation of the appearance of these different variants and the impact of splicing factors and their potential mutations remain unclear.

## 5. Causes of General Splicing Defects in Cancer

AS is a very complex biological mechanism. AS involves a large number of actors allowing a fine regulation of the variant production. Although studies are still needed to better understand and why not modulate this phenomenon, the sequence of events taking place during AS and the molecular partners involved are well described and reported in the review of Ule et al. [107]. Here, we describe the events that can impact AS in tumors (Figure 4).

### 5.1. Mutations of Spliceosome Actors

It should be noted that a germline mutation of one of the spliceosome actors is not viable. This underlines the biological importance of the molecular phenomenon of splicing. Nevertheless, mutations are found in some cancers, such as myelodysplastic syndrome (MDS), which is characterized by inefficient hematopoiesis resulting in cytopenia. In half of the patients with this condition, genes encoding the spliceosome actors SF3B1, SRSF2, U2AF1, and ZRSR2 are mutated in a mutually exclusive way [108]. SF3B1 (splicing factor 3b subunit 1) has been shown to be important for the stabilization of the spliceosome U2 subunit at the pre-mRNA branching site [109]. When the latter is mutated and inoperative, this results in 3’-splice site recognition defects, which result in the appearance of numerous aberrant transcripts comprising premature stop codons within the malignant myeloid cells [110,111]. These transcripts are then degraded via the NMD, which considerably reduces the expression of genes targeted by SF3B1. One of the target genes of SF3B1 is the mitochondrial iron transporter ABCB7 (ATP binding cassette subfamily B member 7). There is then an important accumulation of iron at the mitochondrial level. The latter are often grouped around the nucleus, which forms a so-called sideroblastic crown. Iron stored at the mitochondrial level is then no longer available for hemoglobin synthesis.

### 5.2. Splice Factor Mutations

Splicing factors are essential in the regulation of splicing. Their mutation in the context of cancer can precipitate the transformation of a healthy cell into a cancer cell, or worsen the situation if the mutation is acquired during the development of cancer. The splicing factors are able to bind on more than a hundred genes, and the modification of their expression or their mutation leads to large-scale transcriptomic disturbances, disrupting the biology of the affected cells. For example, in myelodysplastic syndromes, the SRSF2 (serine and arginine-rich splicing factor 2) splicing factor is frequently mutated. This leads to the appearance of splicing aberrations for more than a hundred genes, one of which is of great importance, EZH2 [112]. The latter is involved in the negative regulation of the transcription of many genes by methylation. Thus, many previously extinguished genes are now expressed in an uncontrolled manner. The hematopoietic cells mutated for SRSF2 hold a version of EZH2 including an exon deleterious for its methyltransferase activity. HnRNPs splice factors are also involved in tumor progression. In fact, overexpression of hnRNPA1 in hepatocarcinomas has been shown to be associated with the appearance of metastases via the increase in the v6 isoform of CD44 [113]. In addition, the action of splicing factors may also be disrupted by signaling pathways that may be over-activated in some cancers. Since SR proteins are mainly regulated by phosphorylation, their over-activation can be problematic. SRSF1 is responsible for the inclusion of a caspase-9 exon that results in either caspase-9a or caspase-9b. As the activity of SRSF1 is mainly regulated by AKT, a kinase that is constitutionally active in many cancers such as lung cancer, this leads to a malfunction of caspase-9 which leads to an increased resistance of tumor cells to apoptosis [114]. Many other splicing factors are affected in many cancers and tend to become potential new therapeutic targets.

### 5.3. Impact of Epigenetics on Alternative Splicing and Cancer

Many studies show that epigenetics is globally deregulated in many cancers [115]. The same is true for AS, but very few studies link these two phenomena to the potential consequences of their relationship [116]. In this context, it has recently been shown that there is a link between these two molecular phenomena in colon cancer [117]. SETD2 (SET domain-containing 2), a methyltransferase protein that is often mutated in many cancers, is responsible for the methylation of H3K36 histone. Unlike the methylation of H3K9 and H3K27, this one is permissive and allows the expression of targeted genes. The expression of SETD2 and the methylation of H3K36 are decreased in human colorectal cancer compared to healthy tissues. The decrease in methylation of this histone is superimposed on an increase in alternative aberrant isoforms such as DVL2, a key player in the Wnt signaling pathway. The explanation stems from the fact that, in normal conditions, the permissive methylation of the gene coding for DVL2 leads to a faster transcriptional elongation of RNA polymerase II. Therefore, intron 2 of this gene containing a premature STOP codon is then included in the mRNA, which later causes NMD degradation. When SETD2 is mutated, this is no longer the case because the transcriptional elongation is slower. The intron is then excluded by splicing, and DVL2 can then participate in the Wnt pathway that encourages the phenomenon of cancer transformation.

### 5.4. Influence of TGF-β in Alternative Splicing

It is now clearly recognized that if TGF-β has an anti-proliferative effect on normal cells, it promotes the epithelial-mesenchymal transition of the cancer cells and accentuates the appearance of metastases. This effect notably passes via Smad3, one of the transcription factors activated by the TGF-β pathway. Indeed, it is able to interact with a protein that can bind to the RNA, PCBP1 (poly (RC) binding protein 1). In doing so, this complex can then interact with the variable regions of the CD44 pre-mRNA (described above) and prevent the fixation of the molecular machinery spliceosome. As a result, the appearance of the standard form of CD44, CD44s, which in this context increases the migratory power of cancer cells [118].

Next to the TGF-β signaling pathway mobilizing Smads is the TAK1 kinase. Through its activating action on JNK and p38, it also plays a critical role in the apoptotic functions and EMT caused by TGF-β [119]. TAK1 is also involved in the activation of NFκB, another EMT player, but also cell survival [120,121]. The gene encoding TAK1 comprises 17 exons whose exons 12 and 16 are variable, giving rise to four isoforms. The exclusion of exon 12 from TAK1 has been shown to occur predominantly during EMT, but the link between these two states of affairs has long been unclear (Shapiro et al., 2011). Recently, it has been demonstrated that the exclusion of exon 12 from TAK1 during TGF-β exposure is again caused by the Smad3/PCBP1 complex which is assisted by another RNA-binding protein, Rbfox2 (RNA-binding complex containing Fox-1 homolog 2). It was also found that this isoform losing exon 12, otherwise called TAK1ΔE12, favored EMT through the activation of NFκB. It has also been observed that the action of the complete form of TAK1, TAK1FL, only concerned apoptosis [122]. In addition, it was noted that it was possible to prevent the exclusion of exon 12 through the use of oligonucleotides, thus limiting TGF-β-induced EMT. Finally, TGF-β, when present in the tumor microenvironment, can act on tumor-infiltrating cells, whose lymphocytes. For example, a new isoform of the key transcription factor of Th1 cells, IRF1, was recently described. Indeed, in TGF-β containing environment, IRF1 undergoes an AS giving rise to a new isoform with antagonist functions. This isoform is able to decrease the anti-tumor properties of Th1 cells [50] showing that AS can modify the activity of immune cells.

### 5.5. Metabolism and Hypoxia

Suppressive functions of Treg cells are closely dependent on glycolysis, which controls Foxp3 splicing via the glycolytic enzyme enolase-1 inducing an exon 2-containing variant (Foxp3-E2). Foxp3-E2 disrupts the suppressive activity of Treg cells particularly in human autoimmune diseases, such as multiple sclerosis and type 1 diabetes. Observations showed that the expression of this variant is associated with an alteration of the glycolysis and signaling via interleukin-2 [123]. This link between glycolysis and AS of Foxp3 reveals a new mechanism for regulating the induction and functions of Treg cells with perspectives in human health, particularly in the context of autoimmune diseases.

Disruption of the maintenance of oxygen homeostasis leads to hypoxia. This stress plays an important role in several pathophysiological conditions including cancer. Hypoxic tissues set up adaptation mechanisms that involve a spectacular modulation of gene expression, post-transcriptional, and post-translational modifications of gene expression. AS is one of the responses to hypoxia, particularly in the tumor microenvironment. The effects of hypoxia on the general splicing machinery include the deregulation of splicing factor expression (SRSF1, SRSF2, SRSF3, SAM68, HuR, hnRNPA1, hnRNPM, PRPF40B, and RBM4) or protein kinases (CLK1 and SRPK1) that promote hyperphosphorylation and splice factor activity. Thus, it alters the intracellular location of splice factors and the ability to interact with other proteins and pre-mRNA [124,125,126,127,128]. For example, hypoxia promotes the production of a constitutively active variant of EGFR lacking exons 2 to 7 (EGFRvIII) that correlates with a poor prognosis in several types of tumors including glioblastoma [129]. In another example, hypoxia induces a splice variant of the neurotrophin tyrosine kinase type 1 receptor (TrkA). TrkA is a nerve growth factor (NGF) receptor that plays an essential role in the development of the nervous system. Hypoxia-induced TrkA AS produces a constitutively active form, TrkAIII, of the receptor lacking exons 6, 7, and 9, with strong oncogenic properties [130]. KRAS is another proto-oncogene that undergoes AS of its last exon to generate two almost identical isoforms, KRAS-4A and KRAS-4B, the latter of which lakes exon 4a. When the *kras* gene is mutated, the two isoforms exhibit oncogenic properties. Yet, they behave differently in many aspects of cell biology. For example, KRAS-4B is expressed ubiquitously, while KRAS-4A is mainly limited to tissues of endodermal organs [131]. KRAS-4A promotes apoptosis while KRAS-4B is anti-apoptotic [132]. Tumor hypoxia has been identified as being involved in metastasis formation and has been shown to regulate several invasion-related AS events. For example, hypoxia favors the exclusion of exon 11a from hMENA, an actin regulator, generating a pro-metastatic and pro-invasive splice variant [133]. Another example is the AS of CD44 which promotes migration and invasion processes. Among the 20 exons of the CD44 gene, 10 can undergo AS [134]. In hypoxic regions of breast cancer samples, an elevated expression of CD44 was observed and HIF1a appears to upregulate CD44 variants containing exons V6 and V7/8 [135]. Hypoxia-induced AS is essential and also participates in angiogenesis. Thus, the hypoxia-induced AS promotes a pro-angiogenic AS balance of VEGFA165a, at the expense of anti-angiogenic VEGFA165b.

## 6. Conclusions

Identifying potential targets for AS may provide a basis for the development of new methods of cancer diagnosis and treatments. AS is a regulatory mechanism of cellular functions that can be hijacked in tumor cells. This mechanism also takes place in immune cells, whether under physiological conditions as well as in a tumor microenvironment. The study of splicing events, their regulation as well as their products could lead to a better understanding of this phenomenon under physiological and pathological conditions, and to the identification of new potential therapeutic targets either to directly target tumor cells, angiogenesis, or to restore anti-tumor immunity.

Compounds that affect the overall efficiency of AS have been identified [136]. Among them are Spliceostatin, Sudemycins, and FD-895, and their parent molecules that all act directly on the spliceosome. Isogingetin, another splicing inhibitor, works by preventing the recruitment of U4/U5/U6 snRNPs. In addition, several small molecules modify the activity of splicing factors, for example by targeting their regulatory kinases. Sulfonamides, a class of anticancer drugs, have been shown to act by reducing the expression of the splice factor RBM39 thanks to a new mode of targeted proteasomal degradation [137]. Spinraza and Exondys 51 are also two drugs approved by the FDA in the treatment of spinal atrophy and Duchenne myopathy and were marketed to modify poor quality splicing events. Finally, the use of oligonucleotides to manipulate AS can be of interest in cells bearing splicing alterations.

The impact of AS in the immune response remains an emerging subject that could provide many answers to the fields of investigations in immunology. Nevertheless, if this mechanism is well described in tumor cells, it is still little explored in immune cells. Understanding the AS events that can occur in immune cells present in the tumor microenvironment would allow them to be better used in anti-cancer treatment by immunotherapies. In this field, the use of neoepitopes derived from mRNA splicing as potential targets for anti-cancer cell-based immunotherapeutic approaches and/or vaccination could allow more patients to benefit from these treatments.

Further studies to characterize altered splicing events and splicing factors at the cancer phenotype may be very important for future targeted therapeutic approaches. In parallel, identification of aberrant proteins that could be recurrently generated in the presence of anti-cancer drugs could provide information on the development of resistance to these treatments. Thus, a better understanding of AS in cancer is likely to reveal a wide range of new therapeutic approaches.

## Figures and Tables

**Figure 1 cancers-14-01726-f001:**
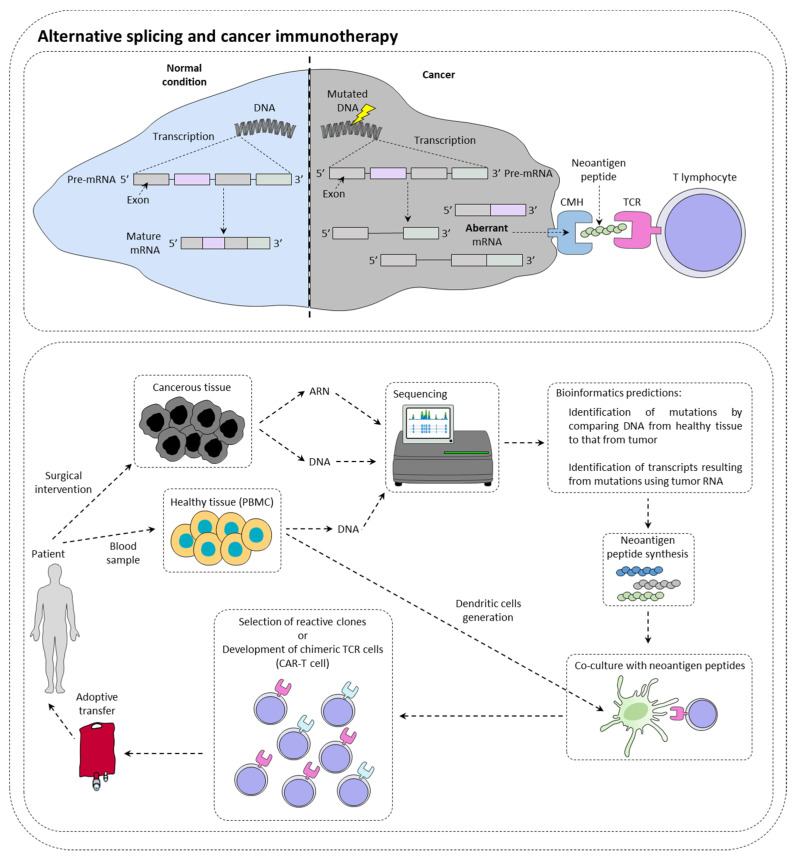
Alternative splicing and cancer immunotherapy. Alternative splicing could have a direct impact on immunotherapy response by creating neoantigen peptide that could be recognized by immune cells. This is currently under investigation for the production of CAR-T cells.

**Figure 2 cancers-14-01726-f002:**
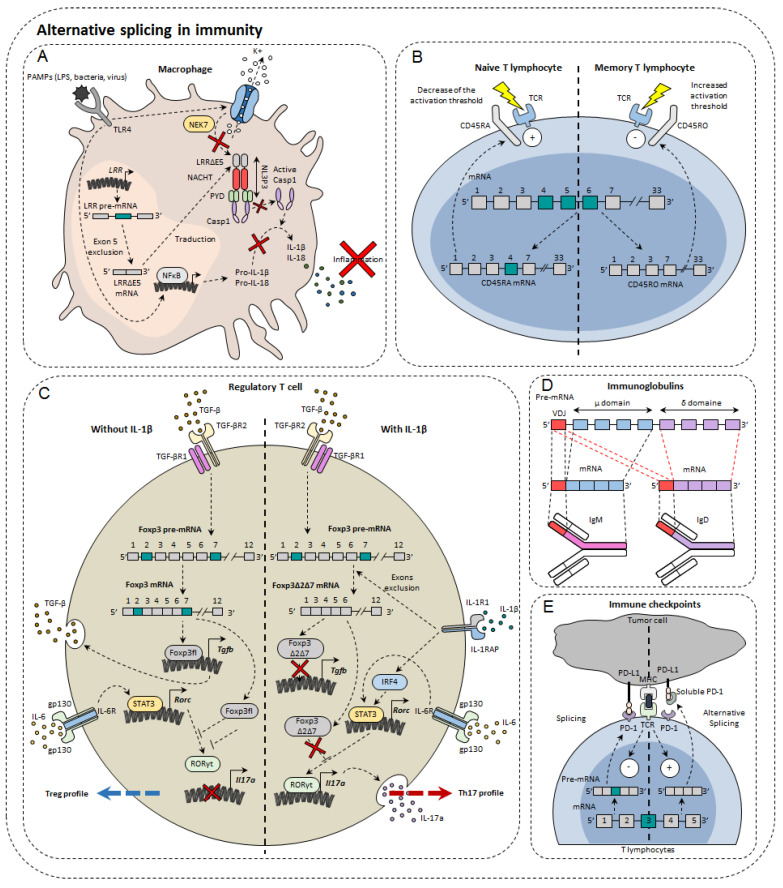
Alternative splicing in immunity. Alternative splicing has a direct impact on normal immune cells such as macrophages (**A**), T lymphocytes (**B**), regulatory T cells (**C**), immunoglobulin diversity (**D**), and immune checkpoints (**E**).

**Figure 3 cancers-14-01726-f003:**
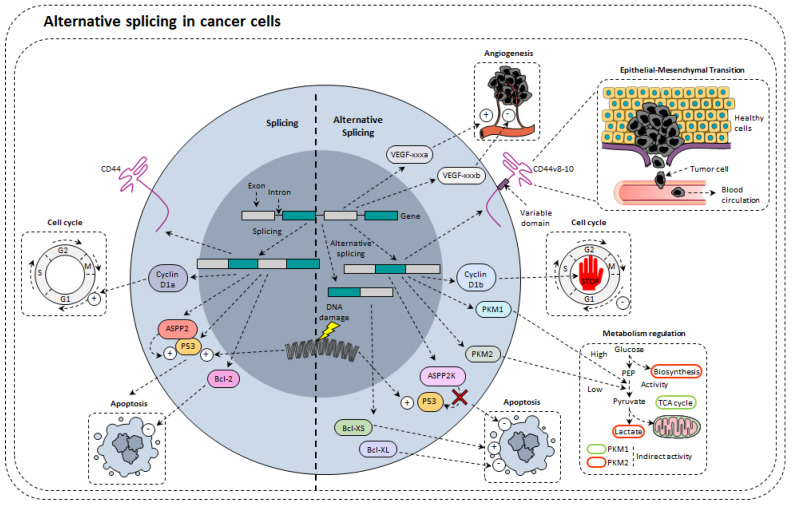
Alternative splicing in cancer cells. Impact of splicing (left part) and alternative splicing (right part) in a cancer cell.

**Figure 4 cancers-14-01726-f004:**
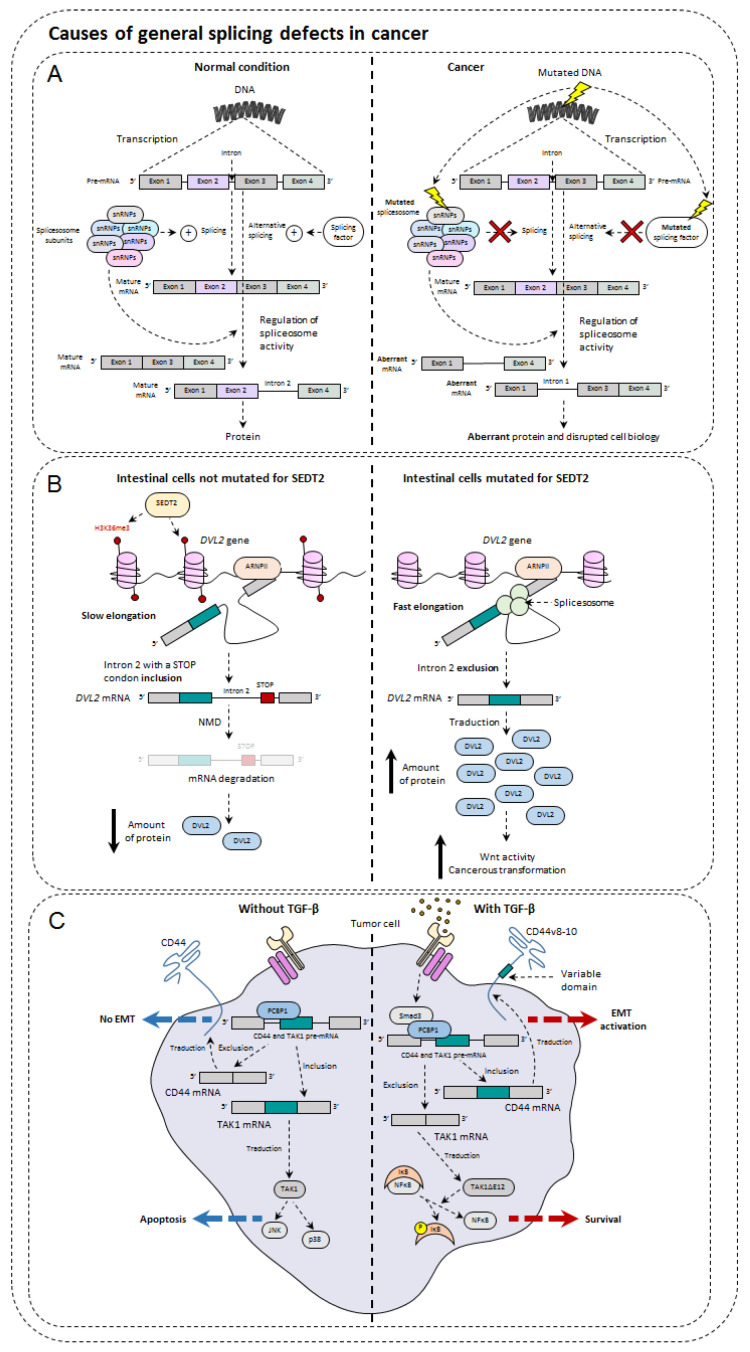
Causes of general splicing defects in cancer. (**A**) Impact of somatic DNA mutation on splicing in cancer cells. (**B**) Impact of SEDT2 mutation on DVL2 mRNA in intestinal cells. (**C**) Role of TGF-β on AS in cancer cells.

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
