# Peer review of "Alternative Splicing in Cancer and Immune Cells"

_cancers, 2022, doi:10.3390/cancers14071726_

Round 1
Reviewer 1 Report
In this review, A Bernard and colleagues adress the issue of alternative splicing in cancer and immune cells.
There is no difined strategy of searching databases, no claim for exhaustivity.
The scope is immense : splicing and cancer has been the object of thousands of reviews before (>3240 in pubmed). The authors mainly give selected examples of alternative splicing in cancer with
- Tumor suppressors (TP53, ASPP2, IRF1)
- Proliferation : cycD1
- Apoptosis (BCL2, BCL2L1, Casapases)
- Angiogenesis (VEGF)
- Metabolism (PKM1/2, ASCL)
- EMT (CD44)
These examples are described side by side, without a clear link between these events, neither functional nor mechanistic. The choice of these specific splicing modifications in cancers (among hundreds of others that have been described) is not justified. It reads more like the proceedings of a meeting with the presentation of various unrelated studies than a real review with a focus or a point of view. Moreover, the very large scope of the review prevents from addressing each aspect with the necessary background, or prompts the use of sometimes scientifically unsound « shortcuts » and even sometimes incomprehensible sentences.
A few examples include :
- « This phenomenon is associated with the nuclear export of P53 in cells expression P53 and P47. »
- « In addition, P47 promotes the mono-ubiquitination of P53, which leads to nuclear export of P53 as well as protein against Mdm2 »
- « In the context of leukemia, it has been demonstrated that the chromosomal region containing the coding portion of IRF1 was deleted [14] » : this sentence is misleading as it would mean that IRF1 deletion is a recognized cause of leukemia while the cited paper mentions that for only a subset of patients. More importantly, variations in IRF1 splicing are described in this paper, which are not mentioned here
- « Binding with nucleophosmin, a factor catalyzing ribosome assembly, also leads to leukemia and MDS loss of IRF1 tumor suppressor function »
A second part reports, again, a number of splicing variations observed in the immune system, either by cell type or by specific function (antigen receptors, transcription factors, immune checkpoints). Again, we have a collection of situations where splicing has been shown to play a role, without a clear link between them or a reason for chosing these specific examples.
A third part rapidely addressees some of the mechanisms at play in splicing anomalies in cancers, from splicing factor mutations to epigenetics, hypoxia and TGF beta signaling.
Finally, a last chapter deals with the consequences of splicing variations on immunotherapy, either by alteration of the targets or by generation of specific neo-antigens.
There is no real link between these chapters besides the fact that they concern splicing, cancer and the immune system.
Even though a number of interesting observations are gathered in this manuscript, they really read like a catalogue of splicing events studied in various circumstances from cancers to immune cells, sometimes presented with mechanistic elements, but no integrative picture. This is likely due to the too large scope of the study. I think the authors should focus their review on a specific point, for example splicing modulation in immunotherapy and give comprehensive and structured insights into this fascinating emerging field.
Author Response
We thank the Reviewer for the time she/he spent to read the manuscript and for her/his comments.
We agree with her/him that the scope is huge. We first tried to modify the manuscript to avoid misleading sentences. Secondly, we put the part on ‘’Alternative splicing and immunotherapy’’ at the beginning of the manuscript. We also added new sentences to make a link between each part.
We hope that these modifications could lead to a better manuscript.

Reviewer 2 Report
The authors summarize a variety of examples of alternative splicing in cancer and immune cells. The review is well written and clear. However, there is too much of mixing in both concepts (cancer in general and immune cells). My comments are below:
- The review should be re-organized. The authors should select the main topic of the review (cancer or immunome). What is less known and of interest is the correlation of alternative splicing with immune cells. This seems to be very novel. Although interconnection exists between cancer in general and immune cells, the contexts need to be separated. The goal of the review needs to be focused.
- Figures are nice but fonts are very small. Font needs to be increased in size by at least two sizes.
- Please expand on immunotherapy and alternative splicing.
Author Response
We warmly thank the Reviewer for her/his constructive comments. In order to answer her/his comments, we decided to start the review with an expanded ‘’Alternative splicing and cancer immunotherapy’’ part. Then, the part on impact of alternative splicing in immune cells followed. The alternative splicing in cancer cells part was put at the end of the manuscript. Specific sentences to link the different parts were added.
Concerning figures, we tried to increase the fonts in order to facilitate the read of the figures.
We hope that these modifications will meet the requirement of the Reviewer for a publication of our review.

Reviewer 3 Report
cancers-1601742
Alternative splicing in cancer and immune cells
Antoine Bernard, Romain Boidot and Frédérique Végran
In this manuscript the authors describe alternative splicing in both normal and cancer cells. They also show their view from the context of immune cells.
This is a well-written and organized review article that has useful information to the readers. I have a few suggestions and comments described below.
- It would be nicer if you could have Tables that compile alternatively spliced genes and splicing regulators involved in.
- I could not find the difference between Figure 2 and 3. Could you please make it clear?
- In page 10, the title of section 4.1 may lack `f` for the word ‘factors’, or do the authors really mean ‘actors’?
Author Response
We warmly thank the Reviewer for her/his comments. We apologize for the mistake in Figures 2 and 3. We made a mistake when we uploaded the figures on the website. The correction was made in the new version of the manuscript.
Concerning the Tables, we did not respond to this point by following academic editor notes.
We hope that the modifications made will meet the requirement for an acceptance of the manuscript.

Round 2
Reviewer 1 Report
The authors provide a revised manuscript in which the order of the chapters has been changed an a few intorductory senstences have been added. This sometimes helps make the manuscript easier to read. However, the issues mentioned in the intial review are sitll present : this is a way too long article trying to embrace too large an area. English still needs improvements.
For example, "2.3. Alternative splicing can limite (sic) Inhibitor of Checkpoint Inhibitors response" makes me wonder whether anyone really proofread before submission...
In short, the remarks of the first review have barely been taken into account. I would be glad to read a much shorter review article that would focus on splicing and immunotherapy, would delete the part about splicing and cancers which is redundant with hundreds of other, better written reviews. This would probably allow authors to correctly proofread the manuscript and propose an article that would be peasant to read.
Author Response
The authors provide a revised manuscript in which the order of the chapters has been changed an a few intorductory senstences have been added. This sometimes helps make the manuscript easier to read. However, the issues mentioned in the intial review are sitll present : this is a way too long article trying to embrace too large an area. English still needs improvements.
For example, "2.3. Alternative splicing can limite (sic) Inhibitor of Checkpoint Inhibitors response" makes me wonder whether anyone really proofread before submission...
We apologize for these problems. We have tried to read the manuscript carefully to correct typos and other kinds of errors in this new version.
In short, the remarks of the first review have barely been taken into account. I would be glad to read a much shorter review article that would focus on splicing and immunotherapy, would delete the part about splicing and cancers which is redundant with hundreds of other, better written reviews. This would probably allow authors to correctly proofread the manuscript and propose an article that would be peasant to read.
We agree that we barely took into account the remarks of Reviewer 1. This is due to the remarks of Reviewers 2 and 3, but also to those of the Academic Editor. Indeed, the remarks of the Reviewer 1 were in contradiction with the remarks of the other actors of the review process. Because of this contradiction, we were note able to take in account all the remarks and decided to follow the concordant remarks.
Reviewer 2 Report
The authors have reorganized the review pretty well. This version has improved and it is well readable.
Author Response
We warmly thank the Reviewer for her/his remarks and for her/his decision.